# Convolutional Neural Networks with Intra-layer Recurrent Connections for Scene Labeling

**Ming Liang**  **Xiaolin Hu**  **Bo Zhang**

Tsinghua National Laboratory for Information Science and Technology (TNList)
Department of Computer Science and Technology
Center for Brain-Inspired Computing Research (CBICR)
Tsinghua University, Beijing 100084, China
liangm07@mails.tsinghua.edu.cn, {xlhu,dcszb}@tsinghua.edu.cn

## Abstract

Scene labeling is a challenging computer vision task. It requires the use of both local discriminative features and global context information. We adopt a deep recurrent convolutional neural network (RCNN) for this task, which is originally proposed for object recognition. Different from traditional convolutional neural networks (CNN), this model has intra-layer recurrent connections in the convolutional layers. Therefore each convolutional layer becomes a two-dimensional recurrent neural network. The units receive constant feed-forward inputs from the previous layer and recurrent inputs from their neighborhoods. While recurrent iterations proceed, the region of context captured by each unit expands. In this way, feature extraction and context modulation are seamlessly integrated, which is different from typical methods that entail separate modules for the two steps. To further utilize the context, a multi-scale RCNN is proposed. Over two benchmark datasets, Standford Background and Sift Flow, the model outperforms many state-of-the-art models in accuracy and efficiency.

## 1 Introduction

Scene labeling (or scene parsing) is an important step towards high-level image interpretation. It aims at fully parsing the input image by labeling the semantic category of each pixel. Compared with image classification, scene labeling is more challenging as it simultaneously solves both segmentation and recognition. The typical approach for scene labeling consists of two steps. First, extract local handcrafted features [6, 15, 26, 23, 27]. Second, integrate context information using probabilistic graphical models [6, 5, 18] or other techniques [24, 21]. In recent years, motivated by the success of deep neural networks in learning visual representations, CNN [12] is incorporated into this framework for feature extraction. However, since CNN does not have an explicit mechanism to modulate its features with context, to achieve better results, other methods such as conditional random field (CRF) [5] and recursive parsing tree [21] are still needed to integrate the context information. It would be interesting to have a neural network capable of performing scene labeling in an end-to-end manner.

A natural way to incorporate context modulation in neural networks is to introduce recurrent connections. This has been extensively studied in sequence learning tasks such as online handwriting recognition [8], speech recognition [9] and machine translation [25]. The sequential data has strong correlations along the time axis. Recurrent neural networks (RNN) are suitable for these tasks because the long-range context information can be captured by a fixed number of recurrent weights. Treating scene labeling as a two-dimensional variant of sequence learning, RNN can also be applied, but the studies are relatively scarce. Recently, a recurrent CNN (RCNN) in which the output of the top layer of a CNN is integrated with the input in the bottom is successfully applied to scene labeling

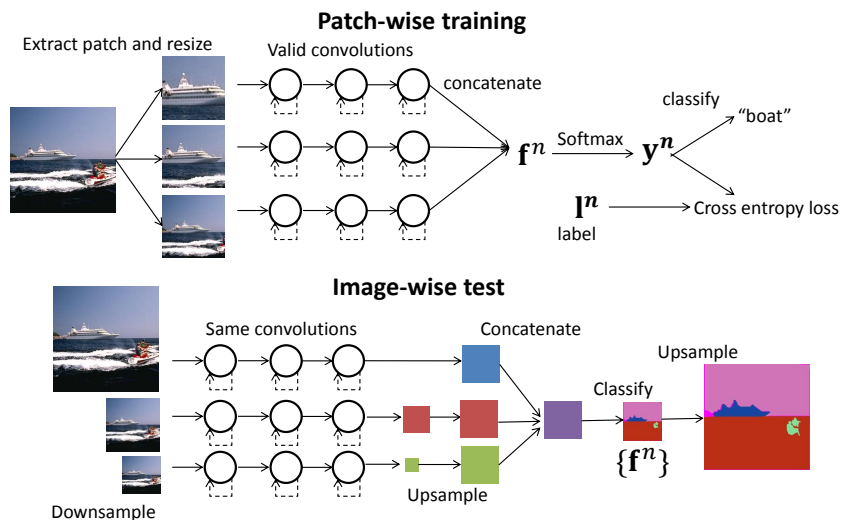

Figure 1: Training and testing processes of multi-scale RCNN for scene labeling. Solid lines denote feed-forward connections and dotted lines denote recurrent connections.

[19]. Without the aid of extra preprocessing or post-processing techniques, it achieves competitive results. This type of recurrent connections captures both local and global information for labeling a pixel, but it achieves this goal indirectly as it does not model the relationship between pixels (or the corresponding units in the hidden layers of CNN) in the 2D space explicitly. To achieve the goal directly, recurrent connections are required to be between units within layers. This type of RCNN has been proposed in [14], but there it is used for object recognition. It is unknown if it is useful for scene labeling, a more challenging task. This motivates the present work.

A prominent structural property of RCNN is that feed-forward and recurrent connections co-exist in multiple layers. This property enables the seamless integration of feature extraction and context modulation in multiple levels of representation. In other words, an RCNN can be seen as a deep RNN which is able to encode the multi-level context dependency. Therefore we expect RCNN to be competent for scene labeling.

Multi-scale is another technique for capturing both local and global information for scene labeling [5]. Therefore we adopt a multi-scale RCNN [14]. An RCNN is used for each scale. See Figure 1 for its overall architecture. The networks in different scales have exactly the same structure and weights. The outputs of all networks are concatenated and input to a softmax layer. The model operates in an end-to-end fashion, and does not need any preprocessing or post-processing techniques.

## 2   Related Work

Many models, either non-parametric [15, 27, 3, 23, 26] or parametric [6, 13, 18], have been proposed for scene labeling. A comprehensive review is beyond the scope of this paper. Below we briefly review the neural network models for scene labeling.

In [5], a multi-scale CNN is used to extract local features for scene labeling. The weights are shared among the CNNs for all scales to keep the number of parameters small. However, the multi-scale scheme alone has no explicit mechanism to ensure the consistency of neighboring pixels' labels. Some post-processing techniques, such as superpixels and CRF, are shown to significantly improve the performance of multi-scale CNN. In [1], CNN features are combined with a fully connected CRF for more accurate segmentations. In both models [5, 1] CNN and CRF are trained in separated stages. In [29] CRF is reformulated and implemented as an RNN, which can be jointly trained with CNN by back-propagation (BP) algorithm.

In [24], a recursive neural network is used to learn a mapping from visual features to the semantic space, which is then used to determine the labels of pixels. In [21], a recursive context propagation

network (rCPN) is proposed to better make use of the global context information. The rCPN is fed a superpixel representation of CNN features. Through a parsing tree, the rCPN recursively aggregates context information from all superpixels and then disseminates it to each superpixel. Although recursive neural network is related to RNN as they both use weight sharing between different layers, they have significant structural difference. The former has a single path from the input layer to the output layer while the latter has multiple paths [14]. As will be shown in Section 4, this difference has great influence on the performance in scene labeling.

To the best of our knowledge, the first end-to-end neural network model for scene labeling refers to the deep CNN proposed in [7]. The model is trained by a supervised greedy learning strategy. In [19], another end-to-end model is proposed. Top-down recurrent connections are incorporated into a CNN to capture context information. In the first recurrent iteration, the CNN receives a raw patch and outputs a predicted label map (downsampled due to pooling). In other iterations, the CNN receives both a downsampled patch and the label map predicted in the previous iteration and then outputs a new predicted label map. Compared with the models in [5, 21], this approach is simple and elegant but its performance is not the best on some benchmark datasets. It is noted that both models in [14] and [19] are called RCNN. For convenience, in what follows, if not specified, RCNN refers to the model in [14].

## 3 Model

### 3.1 RCNN

The key module of the RCNN is the RCL. A generic RNN with feed-forward input $\mathbf{u}(t)$, internal state $\mathbf{x}(t)$ and parameters $\theta$ can be described by:

$$\mathbf{x}(t) = \mathcal{F}(\mathbf{u}(t), \mathbf{x}(t-1), \theta) \tag{1}$$

where $\mathcal{F}$ is the function describing the dynamic behavior of RNN.

The RCL introduces recurrent connections into a convolutional layer (see Figure 2*A* for an illustration). It can be regarded as a special two-dimensional RNN, whose feed-forward and recurrent computations both take the form of convolution.

$$x_{ijk}(t) = \sigma\left((\mathbf{w}_k^f)^\top \mathbf{u}^{(i,j)}(t) + (\mathbf{w}_k^r)^\top \mathbf{x}^{(i,j)}(t-1) + b_k\right) \tag{2}$$

where $\mathbf{u}^{(i,j)}$ and $\mathbf{x}^{(i,j)}$ are vectorized square patches centered at $(i, j)$ of the feature maps of the previous layer and the current layer, $\mathbf{w}_k^f$ and $\mathbf{w}_k^r$ are the weights of feed-forward and recurrent connections for the $k$th feature map, and $b_k$ is the $k$th element of the bias. $\sigma$ used in this paper is composed of two functions $\sigma(z_{ijk}) = h(g(z_{ijk}))$, where $g$ is the widely used rectified linear function $g(z_{ijk}) = \max(z_{ijk}, 0)$, and $h$ is the local response normalization (LRN) [11]:

$$h(g(z_{ijk})) = \frac{g(z_{ijk})}{\left(1 + \frac{\alpha}{L} \sum_{k'=\max(0,k-L/2)}^{\min(K,k+L/2)} (g(z_{ijk'}))^2\right)^\beta} \tag{3}$$

where $K$ is the number of feature maps, $\alpha$ and $\beta$ are constants controlling the amplitude of normalization. The LRN forces the units in the same location to compete for high activities, which mimics the lateral inhibition in the cortex. In our experiments, LRN is found to consistently improve the accuracy, though slightly. Following [11], $\alpha$ and $\beta$ are set to 0.001 and 0.75, respectively. $L$ is set to $K/8 + 1$.

During the training or testing phase, an RCL is unfolded for $T$ time steps into a multi-layer subnetwork. $T$ is a predetermined hyper-parameter. See Figure 2*B* for an example with $T = 3$. The receptive field (RF) of each unit expands with larger $T$, so that more context information is captured. The depth of the subnetwork also increases with larger $T$. In the meantime, the number of parameters is kept constant due to weight sharing.

Let $\mathbf{u}_0$ denote the static input (*e.g.*, an image). The input to the RCL, denoted by $\mathbf{u}(t)$, can take this constant $\mathbf{u}_0$ for all $t$. But here we adopt a more general form:

$$\mathbf{u}(t) = \gamma \mathbf{u}_0 \tag{4}$$

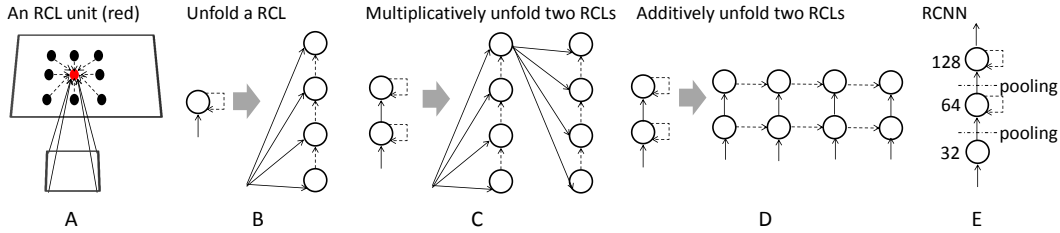

An RCL unit (red)    Unfold a RCL    Multiplicatively unfold two RCLs    Additively unfold two RCLs    RCNN

A                     B               C                                    D                                  E

Figure 2: Illustration of the RCL and RCNN used in this paper. Sold arrows denote feed-forward connections and dotted arrows denote recurrent connections.

where $\gamma \in [0, 1]$ is a discount factor, which determines the tradeoff between the feed-forward component and the recurrent component. When $\gamma = 0$, the feed-forward component is totally discarded after the first iteration. In this case the network behaves like the so-called *recursive convolutional network* [4], in which several convolutional layers have tied weights. There is only one path from input to output. When $\gamma > 0$, the network is a typical RNN. There are multiple paths from input to output (see Figure 2*B*).

RCNN is composed of a stack of RCLs. Between neighboring RCLs there are only feed-forward connections. Max pooling layers are optionally interleaved between RCLs. The total number of recurrent iterations is set to $T$ for all $N$ RCLs. There are two approaches to unfold an RCNN. First, unfold the RCLs one by one, and each RCL is unfolded for $T$ time steps before feeding to the next RCL (see Figure 2*C*). This unfolding approach multiplicatively increases the depth of the network. The largest depth of the network is proportional to $NT$. In the second approach, at each time step the states of all RCLs are updated successively (see Figure 2*D*). The unfolded network has a two-dimensional structure whose $x$ axis is the time step and $y$ axis is the level of layer. This unfolding approach additively increases the depth of the network. The largest depth of the network is proportional to $N + T$.

We adopt the first unfolding approach due to the following advantages. First, it leads to larger effective RF and depth, which are important for the performance of the model. Second, the second approach is more computationally intensive since the feed-forward inputs need to be updated at each time step. However, in the first approach the feed-forward input of each RCL needs to be computed for only once.

### 3.2 Multi-scale RCNN

In natural scenes objects appear in various sizes. To capture this variability, the model should be scale invariant. In [5], a multi-scale CNN is proposed to extract features for scene labeling, in which several CNNs with shared weights are used to process images of different scales. This approach is adopted to construct the multi-scale RCNN (see Figure 1). The original image corresponds to the finest scale. Images of coarser scales are obtained simply by max pooling the original image. The outputs of all RCNNs are concatenated to form the final representation. For pixel $p$, its probability falling into the $c$th semantic category is given by a softmax layer:

$$y_c^p = \frac{\exp\left(\mathbf{w}_c^\top \mathbf{f}^p\right)}{\sum_{c'} \exp\left(\mathbf{w}_{c'}^\top \mathbf{f}^p\right)} \qquad (c = 1, 2, ..., C) \tag{5}$$

where $\mathbf{f}^p$ denotes the concatenated feature vector of pixel $p$, and $w_c$ denotes the weight for the $c$th category.

The loss function is the cross entropy between the predicted probability $y_c^p$ and the true hard label $\hat{y}_c^p$:

$$\mathcal{L} = -\sum_p \sum_c \hat{y}_c^p \log y_c^p \tag{6}$$

where $\hat{y}_c^p = 1$ if pixel $p$ is labeld as $c$ and $\hat{y}_c^p = 0$ otherwise. The model is trained by backpropagation through time (BPTT) [28], that is, unfolding all the RCNNs to feed-forward networks and apply the BP algorithm.

### 3.3 Patch-wise Training and Image-wise Testing

Most neural network models for scene labeling [5, 19, 21] are trained by the patch-wise approach. The training samples are randomly cropped image patches whose labels correspond to the categories of their center pixels. Valid convolutions are used in both feed-forward and recurrent computation. The patch is set to a proper size so that the last feature map has exactly the size of $1 \times 1$. In image-wise training, an image is input to the model and the output has exactly the same size as the image. The loss is the average of all pixels' loss. We have conducted experiments with both training methods, and found that image-wise training seriously suffered from over-fitting. A possible reason is that the pixels in an image have too strong correlations. So patch-wise training is used in all our experiments. In [16], it is suggested that image-wise and patch-wise training are equally effective and the former is faster to converge. But their model is obtained by finetuning the VGG [22] model pretrained on ImageNet [2]. This conclusion may not hold for models trained from scratch.

In the testing phase, the patch-wise approach is time consuming because the patches corresponding to all pixels need to be processed. We therefore use image-wise testing. There are two image-wise testing approaches to obtain dense label maps. The first is the Shift-and-stitch approach [20, 19]. When the predicted label map is downsampled by a factor of $s$, the original image will be shifted and processed for $s^2$ times. At each time, the image is shifted by $(x, y)$ pixels to the right and down. Both $x$ and $y$ take their value from $\{0, 1, 2, \ldots, s - 1\}$, and the shifted image is padded in their left and top borders with zero. The outputs for all shifted images are interleaved so that each pixel has a corresponding prediction. Shift-and-stitch approach needs to process the image for $s^2$ times although it produces the exact prediction as the patch-wise testing. The second approach inputs the entire image to the network and obtains downsampled label map, then simply upsample the map to the same resolution as the input image, using bilinear or other interpolation methods (see Figure 1, bottom). This approach may suffer from the loss of accuracy, but is very efficient. The deconvolutional layer proposed in [16] is adopted for upsampling, which is the backpropagation counterpart of the convolutional layer. The deconvolutional weights are set to simulates the bilinear interpolation. Both of the image-wise testing methods are used in our experiments.

## 4 Experiments

### 4.1 Experimental Settings

Experiments are performed over two benchmark datasets for scene labeling, Sift Flow [15] and Stanford Background [6]. The Sift Flow dataset contains 2688 color images, all of which have the size of $256 \times 256$ pixels. Among them 2488 images are training data, and the remaining 200 images are testing data. There are 33 semantic categories, and the class frequency is highly unbalanced. The Stanford background dataset contains 715 color images, most of them have the size of $320 \times 240$ pixels. Following [6] 5-fold cross validation is used over this dataset. In each fold there are 572 training images and 143 testing images. The pixels have 8 semantic categories and the class frequency is more balanced than the Sift Flow dataset.

In most of our experiments, RCNN has three parameterized layers (Figure 2*E*). The first parameterized layer is a convolutional layer followed by a $2 \times 2$ non-overlapping max pooling layer. This is to reduce the size of feature maps and thus save the computing cost and memory. The other two parameterized layers are RCLs. Another $2 \times 2$ max pooling layer is placed between the two RCLs. The numbers of feature maps in these layers are 32, 64 and 128. The filter size in the first convolutional layer is $7 \times 7$, and the feed-forward and recurrent filters in RCLs are all $3 \times 3$. Three scales of images are used and neighboring scales differed by a factor of 2 in each side of the image.

The models are implemented using Caffe [10]. They are trained using stochastic gradient descent algorithm. For the Sift Flow dataset, the hyper-parameters are determined on a separate validation set. The same set of hyper-parameters is then used for the Stanford Background dataset. Dropout and weight decay are used to prevent over-fitting. Two dropout layers are used, one after the second pooling layer and the other before the concatenation of different scales. The dropout ratio is 0.5 and weight decay coefficient is 0.0001. The base learning rate is 0.001, which is reduced to 0.0001 when the training error enters a plateau. Overall, about ten millions patches have been input to the model during training.

Data augmentation is used in many models [5, 21] for scene labeling to prevent over-fitting. It is a technique to distort the training data with a set of transformations, so that additional data is generated to improve the generalization ability of the models. This technique is only used in Section 4.3 for the sake of fairness in comparison with other models. Augmentation includes horizontal reflection and resizing.

## 4.2 Model Analysis

We empirically analyze the performance of RCNN models for scene labeling on the Sift Flow dataset. The results are shown in Table 1. Two metrics, the per-pixel accuracy (PA) and the average per-class accuracy (CA) are used. PA is the ratio of correctly classified pixels to the total pixels in testing images. CA is the average of all category-wise accuracies. The following results are obtained using the shift-and-stitch testing and without any data augmentation. Note that all models have a multi-scale architecture.

| Model | Patch size | No. Param. | PA (%) | CA (%) |
|---|---|---|---|---|
| RCNN, $\gamma = 1, T = 3$ | 232 | 0.28M | 80.3 | 31.9 |
| RCNN, $\gamma = 1, T = 4$ | 256 | 0.28M | 81.6 | 33.2 |
| RCNN, $\gamma = 1, T = 5$ | 256 | 0.28M | 82.3 | 34.3 |
| RCNN-large, $\gamma = 1, T = 3$ | 256 | 0.65M | **83.4** | **38.9** |
| RCNN, $\gamma = 0, T = 3$ | 232 | 0.28M | 80.5 | 34.2 |
| RCNN, $\gamma = 0, T = 4$ | 256 | 0.28M | 79.9 | 31.4 |
| RCNN, $\gamma = 0, T = 5$ | 256 | 0.28M | 80.4 | 31.7 |
| RCNN-large, $\gamma = 0, T = 3$ | 256 | 0.65M | 78.1 | 29.4 |
| RCNN, $\gamma = 0.25, T = 5$ | 256 | 0.28M | 82.4 | 35.4 |
| RCNN, $\gamma = 0.5, T = 5$ | 256 | 0.28M | 81.8 | 34.7 |
| RCNN, $\gamma = 0.75, T = 5$ | 256 | 0.28M | 82.8 | 35.8 |
| RCNN, no share, $\gamma = 1, T = 5$ | 256 | 0.28M | 81.3 | 33.3 |
| CNN1 | 88 | 0.33M | 74.9 | 24.1 |
| CNN2 | 136 | 0.28M | 78.5 | 28.8 |

Table 1: Model analysis over the Sift Flow dataset. We limit the maximum size of input patch to 256, which is the size of the image in the Sift Flow dataset. This is achieved by replacing the first few *valid* convolutions by *same* convolutions.

First, the influence of $\gamma$ in (4) is investigated. The patch sizes of images for different models are set such that the size of the last feature map is $1 \times 1$. We mainly investigate two specific values $\gamma = 1$ and $\gamma = 0$ with different iteration number $T$. Several other values of $\gamma$ are tested with $T$=5. See Table 1 for details. For RCNN with $\gamma = 1$, the performance monotonously increase with more time steps. This is not the case for RCNN with $\gamma = 0$, with which the network tends to be over-fitting with more iterations. To further investigate this issue, a larger model denoted as RCNN-large is tested. It has four RCLs, and has more parameters and larger depth. With $\gamma = 1$ it achieves a better performance than RCNN. However, the RCNN-large with $\gamma = 0$ obtains worse performance than RCNN. When $\gamma$ is set to other values, 0.25, 0.5 or 0.75, the performance seems better than $\gamma = 1$ but the difference is small.

Second, the influence of weight sharing in recurrent connections is investigated. Another RCNN with $\gamma = 1$ and $T = 5$ is tested. Its recurrent weights in different iterations are not shared anymore, which leads to more parameters than shared ones. But this setting leads to worse accuracy both for PA and CA. A possible reason is that more parameters make the model more prone to over-fitting.

Third, two feed-forward CNNs are constructed for comparison. CNN1 is constructed by removing all recurrent connections from RCNN, and then increasing the numbers of feature maps in each layer from 32, 64 and 128 to 60, 120 and 240, respectively. CNN2 is constructed by removing the recurrent connections and adding two extra convolutional layers. CNN2 had five convolutional layers and the corresponding numbers of feature maps are 32, 64, 64, 128 and 128, respectively. With these settings, the two models have approximately the same number of parameters as RCNN, which is for the sake of fair comparison. The two CNNs are outperformed by the RCNNs by a significant margin. Compared with the RCNN, the topmost units in these two CNNs cover much smaller regions (see the patch size column in Table 1). Note that all convolutionas in these models are performed in "valid" mode. This mode decreases the size of feature maps and as a consequence

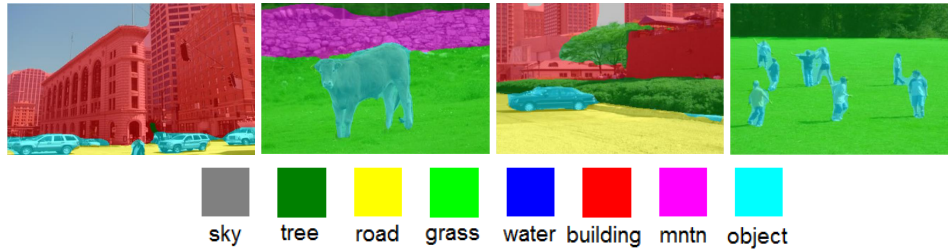

Figure 3: Examples of scene labeling results from the Stanford Background dataset. "mntn" denotes mountains, and "object" denotes foreground objects.

(together with max pooling) increases the RF size of the top units. Since the CNNs have fewer convolutional layers than the time-unfolded RCNNs, their RF sizes of the top units are smaller.

| Model | No. Param. | PA (%) | CA (%) | Time (s) |
|---|---|---|---|---|
| Liu *et al.*[15] | NA | 76.7 | NA | 31 (CPU) |
| Tighe and Lazebnik [27] | NA | 77.0 | 30.1 | 8.4 (CPU) |
| Eigen and Fergus [3] | NA | 77.1 | 32.5 | 16.6 (CPU) |
| Singh and Kosecka [23] | NA | 79.2 | 33.8 | 20 (CPU) |
| Tighe and Lazebnik [26] | NA | 78.6 | 39.2 | $\geq$ 8.4 (CPU) |
| Multi-scale CNN + cover [5] | 0.43 M | 78.5 | 29.6 | NA |
| Multi-scale CNN + cover (balanced) [5] | 0.43 M | 72.3 | 50.8 | NA |
| Top-down RCNN [19] | 0.09 M | 77.7 | 29.8 | NA |
| Multi-scale CNN + rCPN [21] | 0.80 M | 79.6 | 33.6 | 0.37 (GPU) |
| Multi-scale CNN + rCPN (balanced) [21] | 0.80 M | 75.5 | 48.0 | 0.37 (GPU) |
| RCNN | 0.28 M | 83.5 | 35.8 | 0.03 (GPU) |
| RCNN (balanced) | 0.28 M | 79.3 | **57.1** | 0.03 (GPU) |
| RCNN-small | 0.07 M | 81.7 | 32.6 | **0.02** (GPU) |
| RCNN-large | 0.65 M | 84.3 | 41.0 | 0.04 (GPU) |
| FCNN [16] (∗finetuned from VGG model [22]) | 134 M | **85.1** | 51.7 | $\sim$ 0.33 (GPU) |

Table 2: Comparison with the state-of-the-art models over the Sift Flow dataset.

### 4.3 Comparison with the State-of-the-art Models

Next, we compare the results of RCNN and the state-of-the-art models. The RCNN with $\gamma = 1$ and $T = 5$ is used for comparison. The results are obtained using the upsampling testing approach for efficiency. Data augmentation is employed in training because it is used by many other models [5, 21]. The images are only preprocessed by removing the average RGB values computed over training images.

| Model | No. Param. | PA (%) | CA (%) | Time (s) |
|---|---|---|---|---|
| Gould *et al.* [6] | NA | 76.4 | NA | 30 to 60 (CPU) |
| Tighe and Lazebnik [27] | NA | 77.5 | NA | 12 (CPU) |
| Socher *et al.* [24] | NA | 78.1 | NA | NA |
| Eigen and Fergus [3] | NA | 75.3 | 66.5 | 16.6 (CPU) |
| Singh and Kosecka [23] | NA | 74.1 | 62.2 | 20 (CPU) |
| Lempitsky *et al.* [13] | NA | 81.9 | 72.4 | $\geq$ 60 (CPU) |
| Multiscale CNN + CRF [5] | 0.43M | 81.4 | 76.0 | 60.5 (CPU) |
| Top-down RCNN [19] | 0.09M | 80.2 | 69.9 | 10.6 (CPU) |
| Single-scale CNN + rCPN [21] | 0.80M | 81.9 | 73.6 | 0.5 (GPU) |
| Multiscale CNN + rCPN [21] | 0.80M | 81.0 | 78.8 | 0.37 (GPU) |
| Zoom-out [17] | 0.23 M | 82.1 | 77.3 | NA |
| RCNN | 0.28M | **83.1** | 74.8 | **0.03** (GPU) |

Table 3: Comparison with the state-of-the-art models over the Stanford Background dataset.

The results over the Sift Flow dataset are shown in Table 2. Besides the PA and CA, the time for processing an image is also presented. For neural network models, the number of parameters are

shown. When extra training data from other datasets is not used, the RCNN outperforms all other models in terms of the PA metric by a significant margin.

The RCNN has fewer parameters than most of the other neural network models except the top-down RCNN [19]. A small RCNN (RCNN-small) is then constructed by reducing the numbers of feature maps in RCNN to 16, 32 and 64, respectively, so that its total number of parameters is 0.07 million. The PA and CA of the small RCNN are $81.7\%$ and $32.6\%$, respectively, significantly higher than those of the top-down RCNN.

Note that better result over this dataset has been achieved by the fully convolutional network (FCN) [16]. However, FCN is finetuned from the VGG [22] net trained over the 1.2 million images of ImageNet, and has approximately 134 million parameters. Being trained over 2488 images, RCNN is only outperformed by 1.6 percent on PA. This gap can be further reduced by using larger RCNN models. For example, the RCNN-large in Table 1 achieves PA of $84.3\%$ with data augmentation.

The class distribution in the Sift Flow dataset is highly unbalanced, which is harmful to the CA performance. In [5], frequency balance is used so that patches in different classes appear in the same frequency. This operation greatly enhance the CA value. For better comparison, we also test an RCNN with weighted sampling (balanced) so that the rarer classes apprear more frequently. In this case, the RCNN achieves a much higher CA than other methods including FCN, while still keeping a good PA.

The results over the Stanford Background dataset are shown in Table 3. The set of hyper-parameters used for the Sift Flow dataset is adopted without further tuning. Frequency balance is not used. The RCNN again achieves the best PA score, although CA is not the best. Some typical results of RCNN are shown in Figure 3.

On a GTX Titan black GPU, it takes about 0.03 second for the RCNN and 0.02 second for the RCNN-small to process an image. Compared with other models, the efficiency of RCNN is mainly attributed to its end-to-end property. For example, the rCPN model takes much time in obtaining the superpixels.

## 5 Conclusion

A multi-scale recurrent convolutional neural network is used for scene labeling. The model is able to perform local feature extraction and context integration simultaneously in each parameterized layer, therefore particularly fits this application because both local and global information are critical for determining the label of a pixel in an image. This is an end-to-end approach and can be simply trained by the BPTT algorithm. Experimental results over two benchmark datasets demonstrate the effectiveness and efficiency of the model.

## Acknowledgements

We are grateful to the anonymous reviewers for their valuable comments. This work was supported in part by the National Basic Research Program (973 Program) of China under Grant 2012CB316301 and Grant 2013CB329403, in part by the National Natural Science Foundation of China under Grant 61273023, Grant 91420201, and Grant 61332007, in part by the Natural Science Foundation of Beijing under Grant 4132046.

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
