[Reviews · NeurIPS 2015]

Submitted by Assigned_Reviewer_1

This paper presents a recurrent convolutional neural network for semantic image segmentation to encode and take advantage of contextual relationships. The method is basically a combination of [13] where a very similar RCNN is used for object recognition, and [4] from which the multi-scale pipeline is inspired. Thus, technically, the paper is not very novel (Sec. 3.1. is much the same as in previous works - maybe that should be stated more clearly; the combination can be seen as a novelty of course). However, the paper is well executed, very easy and clear to read, largely well written and provides a seemingly fair evaluation to state-of-the-art. Some recent works or interesting evaluations could be added, see below. Results are shown on the Sift Flow and the Stanford Background dataset where the proposed technique outperforms state-of-the-art using a limited number of training data for all approaches (limited by these datasets). Also, parameter ablation studies are conducted to some extend. The method is very efficient.

Detailed comments:

- The related work section could be written more sharply. It is not always crystal clear what the differences are with respect to the closest related works.

- Results on PASCAL VOC could be added as most recent segmentation works evaluate on those datasets.

- l.276 is very vague and should be made more clear.

- Table 1: Why are only \gamma \in {0,1} considered? The paper could provide experiments/plots with varying gammas.

- page 7: The ordering of text/plots/tables should be changed to have less interleaving text/plots/tables.

- The model mentioned in l.387 should be added to the table, maybe with a footnote that it uses different training data or a line break.

- Some further qualitative analysis would be nice if it fits.

- This recent work seems to be missing (the first one provides even stronger results on Stanford Background):

Feedforward semantic segmentation with zoom-out features Mohammadreza Mostajabi, Payman Yadollahpour and Gregory Shakhnarovich Toyota Technological Institute at Chicago

@inproceedings{crfasrnn_arXiv2015,

author = {Shuai Zheng and Sadeep Jayasumana and Bernardino Romera-Paredes and Vibhav Vineet and Zhizhong Su and Dalong Du and Chang Huang and Philip Torr},

title = {Conditional Random Fields as Recurrent Neural Networks},

booktitle = {arXiv:1502.03240}, year = {2015}

}
Summary: The paper is well executed and provides a principled combination of two existing techniques. The results seem convincing, up to some additional studies which would benefit the paper (see below).

Submitted by Assigned_Reviewer_2

This paper presents a pixelwise classification system using recurrent convolutional connections.

The method employs a multiscale convolutional network similar to Farabet et al., but introduces shared weights and additive skip-connections between layers in the networks applied at each scale.

The model is evaluated on Stanford Background and SIFT Flow datasets.

This approach appears to achieve good performance for being trained from scratch, but I think the several aspects could be better explored and evaluated.

* First, the effect of sharing weights between recurrent connections could be compared to the same network with different convolution kernels in each iteration; while this introduces more parameters, it might also stand to increase performance for about the same computational cost (i.e., same number of connections).

This was almost performed, but the CNN2 model has fewer total layers and a smaller RF.

It is also possible the findings for varying gamma (enabling/disabling skip connections) might change under these conditions.

* Second, the effects of N and T could be further expanded upon.

From what I can tell, all the RCNN networks use T=3, except RCNN-large, which uses T=4.

But RCNN-large is evaluated only for N=3.

How does it perform at N=4 and N=5?

* In addition, Table 2 RCNN has a reported performance of 83.5/35.8 PA/CA.

But in Table 1, RCNN-large has what looks like better performance at 83.4/38.9 PA/CA (last line).

Is there a reason the latter wasn't used in Table 2?

* The related work section states that [4] and similar networks rely on postprocessing, e.g. superpixels, to ensure consistency of neighboring pixels labels.

This seems to imply the proposed model does not suffer from this problem, but this is not evaluated in the experiments or with qualitative examples.

No example predictions are shown, so it isn't really clear what the outputs look like qualitatively.

* There are also some other recent works in this area that I think could be discussed or compared with, particularly Chen et al. "Semantic Image Segmentation with Deep Convolutional Nets and Fully Connected CRFs", and Eigen & Fergus "Predicting Depth, Surface Normals and Semantic Labels with a Common Multi-Scale Convolutional Architecture"
Summary: This approach appears to achieve good performance for being trained from scratch, however I think some aspects could be better explored and evaluated.

Submitted by Assigned_Reviewer_3

The authors claim that the effective receptive field of each unit expands with larger T (Line 174 in Page 4). I didn't get this point. As you input an still image to the network, how can the receptive field of each unit get expanded with larger T? Can you clarify on this?
Summary: The paper proposes a recurrent convolutional layer for CNN to improve the performance of scene image parsing. The algorithm looks technically sound and the result looks good.

Submitted by Assigned_Reviewer_4

[this is a light review] Specifically similar approache: - Shuai Zheng; Sadeep Jayasumana; Bernardino Romera-Paredes; Chang Huang; Philip Torr, http://arxiv.org/abs/1502.03240. - [15] should be included in the results table, maybe with a star to denote different/more training data.

Summary: The paper adapts the recurrent neural network approach for object detection [13] to scene labeling/semantic segmentation. The approach outperforms baselines but not state-of-the-art. The authors should relate their work conceptually, quantitatively and w.r.t. the difference in approach more clearly from the semantic segmentation task & datasets and corresponding approaches (see e.g. http://host.robots.ox.ac.uk:8080/leaderboard/displaylb.php?challengeid=11&compid=6 for a list of winning approaches).

Submitted by Assigned_Reviewer_5

This paper presents a method for scene labelling based on Recurrent Convolutional Neural Networks, where the output of a convolutional layer is used as an additional input of the same layer (this is implemented by duplicating the layer several times). The input to the network is the input image plus several downscaled versions of of the input image to exploit contextual information better.

The approach is tested evaluated on two datasets, and compared to previous methods. The accuracy improvement is good, and the computation time impressive as the algorithm can run entirely on the GPU.

I found the paper interesting as it discusses very trendy issues but I have two main concerns: - the text is difficult to follow. For example, the use of the passive form in the first sentence of the last paragraph of the introduction makes it ambiguous. It took me some time to understand (or guess) that the authors meant "we adopt a multi-scale version of RCNN". - while the results are interesting, the contribution is limited, as it is only an application of RCNN (which were already applied to computer vision problems before) to image labeling.

More minor comments: - Section 3: state explicitly that RCL stands for recurrent convolutional layer. Same problem with LRN - just before Eq (3): it should be g(Z_ijkz), not sigma(z_ijk) - is it really worth discussing gamma?

The results with gamma = 0 are not as good as with gamma = 1, which is the "standard" way. This should not be surprising, because if a small gamma was interesting, the network could learn to use large values for w_k^rec
Summary: This paper presents a method for scene labelling based on Recurrent Convolutional Neural Networks. Results are interesting, but the text is difficult to follow and the contribution seems limited.

Author Feedback
Author rebuttal: We thank all the reviewers for their acknowledgements of our work and for the valuable suggestions. The indicated missing references will be added, and the tables and text will be reorganized to make the presentation clearer. Below we address the comments in detail.

To Review_1:
Thanks for your suggestion on the related work section. We will modify it to emphasize the differences with the closest related works.

The results on VOC are not included because the state-of-the-art models on these datasets usually use the ImageNet pre-trained models, such as VGG net or GoogLeNet. But we agree that these results are good complement to the present experiments. If there is enough space, these results will be added.

In L. 276, we mean that data augmentation is not used in model analysis, but is used in comparison with the state-of-the-art models because these models also use data augmentation. We will clarify this point.

We did test some values of gamma between 0 and 1, and some values had led to a little bit better results than gamma=1, but the difference was small. To save space these results were not included. We will add these results by reorganizing the presentation.

The model mentioned in L.387 will be added to Table 2.

Good suggestions for adding qualitative analysis. We will do that.

Thanks for the indication of the missing references.

To Reviewer_3:
We are very sorry for the ambiguity in writing. We will try our best to improve the writing and eliminate these ambiguities.

The current work is a nontrivial application of RCNN. It is worthwhile to show that a model good at object recognition is also good at scene labeling as the two tasks are different. That's also why many other deep learning models originally used for other tasks are adapted for scene labeling (e.g., [4,15,19]). Two advantages of RCNN for scene labeling refer to its simplicity and efficiency, which are believed to be favored by the practitioners working in this field.

Thanks for indicating the missing notations of RCL and LRN and the typo just before Eq (3). They will be corrected.

By comparing different gamma, we intend to emphasize the effect of the multi-path structure. When gamma=0, there is only one path from input to output.

To Reviewer_4:
Thanks for the acknowledgement for our contributions.

To Reviewer_6:
Yes, the input is a still image. The dynamics comes from the recurrent iterations. The RCLs are unfolded through time for T steps. In each step, each unit receives recurrent inputs from its neighboring units, and these inputs bring information in a larger context in the layer below. In other words, in each step, a larger area (receptive field, RF) in the layer below modulates the activity of each unit in the current layer.

To Reviewer_7:
Thanks for indicating the webpage and the reference. An explanation for not having included the VOC results is found in our reply to Reviewer_1. We will modify the paper to clarify the difference of RCNN with other approaches, and add more experimental results if space permits

To Reviewer_8:
Thanks for the suggestion to compare the shared and unshared weights. This is indeed useful to illustrate the effect of weight sharing. Our results showed that changing the shared weights to unshared weights made the score dropped by about 1%. It may be due to the over-fitting with more parameters. We will add these results in the experiment section.

Yes, the findings about gamma may change under different conditions, e.g., when the model is smaller. We'll try different settings.

More iterations lead to larger effective RFs for the units. Because RCNN-large contains N=4 RCLs, and the effective RFs of top units already cover the entire image with T=3 iterations. Further increasing the iterations will not make their effective RFs cover more areas in the image. But RCNN contains only N=2 RCLs, and we can use more iterations. Sorry we made a mistake: the N's in Table 1 should be T's. We guess this typo has caused confusion to you.

Yes, RCNN-large did perform better than RCNN. But to emphasize that RCNN with relatively few parameters could achieve good results, we didn't compare RCNN-large with existing methods in Table 2 because it had relatively more parameters (though still fewer than some existing models). It was only used to show that with more parameters (but not more data) the accuracy of RCNN could approach that of a model pre-trained on ImageNet (see L.387).

Good suggestions for adding qualitative analysis. We will do that.

Thanks for indicating the related references. We will add discussions and comparisons with them.